

# The contribution of estimated dead space fraction to mortality prediction in patients with chronic obstructive pulmonary disease—a new proposal

Ming-Lung Chuang[1,2], Yu Hsun Wang[3] and I-Feng Lin[4]

[1] School of Medicine, Chung Shan Medical University, Taichung, Taiwan
[2] Div. Pulmonary Medicine, Department of Internal Medicine, Chung Shan Medical University Hospital, Taichung, Taiwan
[3] Department of Medical Research, Chung Shan Medical University Hospital, Taichung, Taiwan
[4] Institute of Public Health, National Yang Ming Chiao Tung University, Taipei, Taiwan

Corresponding author
Ming-Lung Chuang,
yuan1007@ms36.hinet.net

## ABSTRACT

**Background:** Mortality due to chronic obstructive pulmonary disease (COPD) is increasing. However, dead space fractions at rest ($V_D/V_{Trest}$) and peak exercise ($V_D/V_{Tpeak}$) and variables affecting survival have not been evaluated. This study aimed to investigate these issues.

**Methods:** This retrospective observational cohort study was conducted from 2010–2020. Patients with COPD who smoked, met the Global Initiatives for Chronic Lung Diseases (GOLD) criteria, had available demographic, complete lung function test (CLFT), medication, acute exacerbation of COPD (AECOPD), Charlson Comorbidity Index, and survival data were enrolled. $V_D/V_{Trest}$ and $V_D/V_{Tpeak}$ were estimated ($estV_D/V_{Trest}$ and $estV_D/V_{Tpeak}$). Univariate and multivariable Cox regression with stepwise variable selection were performed to estimate hazard ratios of all-cause mortality.

**Results:** Overall, 14,910 patients with COPD were obtained from the hospital database, and 456 were analyzed after excluding those without CLFT or meeting the lung function criteria during the follow-up period (median (IQR) 597 (331–934.5) days). Of the 456 subjects, 81% had GOLD stages 2 and 3, highly elevated dead space fractions, mild air-trapping and diffusion impairment. The hospitalized AECOPD rate was $0.60 \pm 2.84$/person/year. Forty-eight subjects (10.5%) died, including 30 with advanced cancer. The incidence density of death was 6.03 per 100 person-years. The crude risk factors for mortality were elevated $estV_D/V_{Trest}$, $estV_D/V_{Tpeak}$, ≥2 hospitalizations for AECOPD, advanced age, body mass index (BMI) <18.5 kg/m$^2$, and cancer (hazard ratios (95% C.I.) from 1.03 [1.00–1.06] to 5.45 [3.04–9.79]). The protective factors were high peak expiratory flow%, adjusted diffusing capacity%, alveolar volume%, and BMI 24–26.9 kg/m$^2$. In stepwise Cox regression analysis, after adjusting for all selected factors except cancer, $estV_D/V_{Trest}$ and BMI <18.5 kg/m$^2$ were risk factors, whereas BMI 24–26.9 kg/m$^2$ was protective. Cancer was the main cause of all-cause mortality in this study; however, $estV_D/V_{Trest}$ and BMI were independent prognostic factors for COPD after excluding cancer.

**Conclusions:** The predictive formula for dead space fraction enables the estimation of $V_D/V_{Trest}$, and the mortality probability formula facilitates the estimation of

COPD mortality. However, the clinical implications should be approached with caution until these formulas have been validated.

# INTRODUCTION

Chronic obstructive pulmonary disease (COPD) is a common lung disease, with a global prevalence of 11.7% (*GOLD Committees, 2022*). The mortality rate associated with COPD continues to increase, even though those associated with other chronic diseases have decreased (*WHO, 2022*). Survival in COPD has been extensively studied using various techniques, including composite indexes (*Celli et al., 2004*; *Leivseth et al., 2013*; *Puhan et al., 2009*), Global Initiatives for Chronic Lung Diseases (GOLD) classification (*Gedebjerg et al., 2018*; *Lee et al., 2019*; *Leivseth et al., 2013*) and indexes using multivariate analysis (*Domingo-Salvany et al., 2002*; *Martinez et al., 2006*; *Puhan et al., 2009*; *Soler-Cataluna et al., 2009*). However, *Gosker et al. (2002)* and *Hurst & Wedzicha (2007)* suggested that COPD survival factors can be stratified in a step-by-step hierarchical fashion according to the conceptual framework between pathogenesis and survival of COPD patients. Accordingly, the survival factors are categorized as follows: (1) primary lung function variables *i.e.* $FEV_1$% (*Leivseth et al., 2013*; *Ou et al., 2014*; *Soriano et al., 2013*), its related variables (*Huang et al., 2018*), diffusing capacity of the lungs for carbon monoxide ($D_LCO$) (*de-Torres et al., 2021*), inspiratory capacity (*Phillips et al., 2022*) and total lung capacity ratio (IC/TLC) (*Aalstad et al., 2018*); primary lung structural variable *i.e.* emphysema (*Martinez et al., 2006*); (2) secondary lung variables *i.e.* hypoxemia and hypercapnia (*Dave et al., 2021*; *Soler-Cataluna et al., 2005*); and (3) tertiary symptoms, exercise intolerance or health status (*Nishimura et al., 2002*; *Oga et al., 2003*), and acute exacerbations (*Soler-Cataluna et al., 2005*, *2009*) caused by deranged lungs. Non-lung factors may also affect COPD survival, including age, sex, BMI, muscle power, cardiac function, and comorbidities (*Divo et al., 2012*).

However, elevated dead space fraction ($V_D/V_T$) is a dysfunctional primary lung variable. The Bohr-Enghoff standard equation has been clearly outlined for its calculation (*Chuang, Hsieh & Lin, 2021*; *Wasserman et al., 2005*). Typically, the $V_D/V_T$ represents physiological $V_D/V_T$. It is more related to clinical characteristics and gas exchange than $FEV_1$ (*Chuang, 2020*; *Chuang, Hsieh & Lin, 2021*). It is also associated with tidal inspiratory flow and operational lung volume at peak exercise, and the diffusing capacity of the lungs for carbon monoxide (DLCO) at rest (*Chuang, 2022*). $V_D/V_T$ has yet to be shown to a predictor of COPD mortality; however, it is related to hypoxemia and hypercapnia, which are in turn related to mortality (*Calverley, 2003*; *Dave et al., 2021*; *Mathews et al., 2020*; *Nizet et al., 2005*). Therefore, the association of $V_D/V_T$ with COPD mortality is speculative. However, to obtain $V_D/V_T$ invasive arterial catheterization must be established, and applying an invasive method in a large-scale study is difficult. Thus, using predictive formulae to

estimate $V_D/V_T$ values at rest (est$V_D/V_{Trest}$) and peak exercise (est$V_D/V_{Tpeak}$) may be a more suitable option. In addition as the causes of death in patients with mild COPD are predominantly cancer and cardiovascular disease (*Berry & Wise, 2010*) and the mortality rate of patients with cancer with metastasis is speculated to be much higher than that of COPD alone, where patients with advanced-stage malignancy of any organ have been excluded in previous studies on COPD outcomes (*Huang et al., 2018*). We hypothesized that adjusting for the relevant factors without cancer, may affect the survival analysis of COPD.

Hence, this study aimed to assess (1) all-cause mortality and (2) the independent contribution of $V_D/V_T$, excluding cancer, to predict mortality in patients with COPD.

## METHODS

Portions of this text were previously published as part of a preprint (https://doi.org/10.21203/rs.3.rs-2383786/v1).

### Study design

This was a retrospective cross-sectional observational hospital-based cohort study. We screened all subjects with COPD who were outpatient clinic patients and patients hospitalized with an exacerbation and had available data on demographics, complete lung function tests, comorbidities, number of hospitalized acute exacerbations of COPD (AECOPD), inhaled medications, and survival from the electronic medical records of our hospital. We then calculated est$V_D/V_{Trest}$ and est$V_D/V_{Tpeak}$ (please refer to $V_D/V_T$ *measurement* section). Survival data were censored and were double-checked against the National Death Index for correctness and to avoid missing data due to those who were lost to follow-up. This study was approved by the Institutional Review Board (IRB) of Chung Shan Medical University Hospital (CS2-21018). All methods were performed in accordance with Declarations of Helsinki. Informed consent was waived from IRB. The data were accessed for research purposes from 23rd February, 2021 to 22nd February, 2022. The authors had access to information that could not identify individual participants during or after data collection.

### Study population

We enrolled patients with COPD according to International Classification of Diseases 9th and 10th Revisions (ICD-9 and ICD-10) code 428 (428.0 to 428.9) (*Silvestre et al., 2018*) or a primary diagnosis of COPD (J44.X) or COPD (J44.X) in combination with acute respiratory failure (J96.X) as a secondary diagnosis between January 01, 2010 and December 31, 2020. Our study population included incident and prevalent patients aged 40 years or older who had visited the hospital's outpatient clinics or been hospitalized. The data included computerized discharge records for all hospital admissions and outpatient visits to specialist clinics or emergency departments. The inclusion criteria were a diagnosis of COPD, cigarette smoker, and available data on complete lung function tests, including spirometry, lung volume measures, and $D_L$CO. A diagnosis of COPD was defined as $FEV_1$/forced vital capacity (FVC) <0.7 with an insignificant bronchodilator

effect according to the GOLD criteria (*GOLD Committees, 2022*). The exclusion criteria were lung diseases other than COPD or COPD with lung cancer and mixed ventilatory defect *i.e.* TLC < 80% predicted. This criterion was applied to avoid potential inclusion of individuals with thoracic diseases characterized by restrictive ventilation defects, such as interstitial disease or suspected rib cage defects. Subjects in whom the single breath dilution method was used for lung volume measures before 2013 were excluded, as the data obtained by this method were significantly different from those obtained by body plethysmography after 2013. Lung volumes measured with these two methods are different in patients with COPD (*O'Donnell et al., 2010*), and body plethysmography is preferred to assess the therapeutic effect in patients with COPD and lung hyperinflation (*Cazzola et al., 2009*).

*Complete pulmonary function tests (PFTs).* Complete PFTs were conducted by trained technicians at the pulmonary function laboratory. The comprehensive PFT included measurements of $FEV_1$, FVC, peak flow rate (PEF), maximum mid-expiratory flow (MMEF), lung volumes, and $D_LCO$. The results were reported at body temperature, ambient atmospheric pressure, and full saturation, utilizing the best readings from three technically satisfactory attempts (*ATS/ERS, 2002*; *Miller et al., 2005a*, *2005b*). The equipment used for these measurements included either the 6200 Autobox DL by SensorMedics in California, USA, or the MasterScreen™ Body by Carefusion in Würzburg, Germany. All lung function data were expressed as % predicted as reported in previous studies (*Chuang & Lin, 2019*). The rationale for this approach was to keep our lung function reports consistent, and thus we did not use Global Lung Function Initiative reference values (*Quanjer et al., 2012*). The currently utilized predicted values in our institute are as follows: race adjustment for $FEV_1$ and FVC was performed employing 90% of the predictive equations established by *Knudson et al. (1976)*. The predicted values for TLC and $D_LCO$ were derived using 85% of the prediction equations developed by *Goldman & Becklake (1959)* and *Burrows et al. (1961)*, respectively. COPD severity was based on spirometry according to GOLD classifications.

*$V_D/V_T$ measurement.* In general, arterial and mixed expired $PCO_2$ have to be measured to obtain $V_D/V_T$, where one is invasive and the other is technically sophisticated. However, $V_D/V_T$ can be estimated using statistical methods (*Chuang, Hsieh & Lin, 2021*, *2022*). The formula for $estV_D/V_{Trest}$ is as follows: 1.046 (±0.279) − 0.0042 × height (±0.0015) + 0.0014 × smoke (±0.0006) + 0.491 × HTR (±0.276) ($r^2$ = 0.33, $p$ = 0.001). Here height is in centimeters, smoke represents cigarette consumption in pack-year, and HTR is the hila-thoracic ratio measured from chest radiography, and the numbers in parenthesis are standard errors. The formula for $estV_D/V_{Tpeak}$ is as follows: 1.1375 (±0.0751) − 0.0333 × SVC (±0.0176) + 0.0045 × $D_LCO$ (±0.0024) + 0.1346 × $V_T/T_{Ipeak}$ (±0.0359) − 0.0037 × $HR_{peak}$ (±0.0006) − 0.0492 × $O_2P_{peak}$ (±0.0067) ($r^2$ = 0.74, $p$ < 0.0001). Here SVC is slow vital capacity in liters, $D_LCO$ is diffusing capacity of the lungs for carbon monoxide in mL/min/mm Hg, $V_T/T_{Ipeak}$ is inspiratory tidal flow at peak exercise in liters/s, $HR_{peak}$ is heart rate at peak exercise in beats/min, and $O_2P_{peak}$ is oxygen pulse at peak exercise in milliliters/beat. As some variables were not available for this study, in this study, $V_D/V_{Trest}$

and $V_D/V_{Tpeak}$ were obtained using simplified versions of the equations using multiple linear regression analysis as follows: $estV_D/V_{Trest} = 1.18449 + 0.00356 \times age - 0.00416 \times height - 0.05676 \times FRC\% - 0.12206 \times FEV_1/FVC$ ($r^2 = 0.28$, $p = 0.01$) and $estV_D/V_{Tpeak} = 0.48428 - 0.00189 \times Weight - 0.07579 \times TLC\% + 0.42374 \times RV/TLC - 0.11938 \times D_LCO\%$ ($r^2 = 0.31$, $p = 0.007$), where FRC% is functional residual capacity % predicted, FVC is forced vital capacity, TLC% is total lung capacity % predicted, RV is residual volume. These formulae were revised from our previous reports using fewer and available variables for simplicity and generalizability (*Chuang, Hsieh & Lin, 2021*; *Chuang, 2022*).

*Inhaled medications.* The inhaled medications were classified into inhaled corticosteroids (ICS), long-acting beta 2 agonists (LABA), long-acting muscarinic antagonists (LAMA), LABA plus ICS, LABA plus LAMA, and LABA+LAMA+ICS groups. Each subject was classified into only one group according to the highest number of combined medications. LABA+ICS has been suggested for moderate to very severe COPD patients in GOLD guidelines since 2013 (*Vestbo et al., 2013*) and has been changed for subjects who have frequent AEs/hospitalization(s), more symptoms, and eosinophil >300/µL since 2020 (*GOLD Committees, 2020*). Coincidentally, the current study population were enrolled between 2013 and 2020.

*Co-morbidities.* We categorized comorbidities according to the Charlson Comorbidity Index (CCI) within the study period before the index date. The scores were stratified into ≤3, 4–6, and >6 or ≤2 and >2, respectively. The chronic pulmonary disease category was not included because it constituted the index disease of our cohort.

*Exacerbations.* To obtain the frequency of hospitalized AECOPDs, we used information from electronic medical records on COPD-related inpatient admissions (ICD-10 codes J41-J44, J96, J12-J18 (pneumonia), or J20-J22 (acute lower respiratory infections)) during the study period. All hospitalized AECOPDs were checked for correctness by the investigators. Only moderate to severe AECOPDs requiring hospital admission were defined as hospitalized AECOPDs, and expressed as per person per year (PPPY).

*Outcomes.* The primary outcome was all-cause mortality. The exact date of death was obtained from the hospital's database. Deaths were confirmed through linkage with the National Death Index. The National Death Index in Taiwan may provide individual mortality data including the date and causes on request under a formal application with local IRB approval. The incidence density of death was expressed as per 100 person-years. Identifying the risk factors for mortality other than cancer was another goal of this study, and therefore subjects with cancer were subsequently excluded (*Huang et al., 2018*).

*Covariates.* Covariates were extracted from medical records database, including age, sex, BMI, CCI scores, number of hospitalized AECOPDs, and prescriptions for inhaled medications. In Taiwan, BMI < 18.5 kg/m$^2$ is underweight, 18.5–23.9 kg/m$^2$ is normal, 24–26.9 kg/m$^2$ is overweight, and >27 kg/m$^2$ is obesity (*MOHW, 2018*).

## Statistical analysis

The raw data supporting the conclusions of this article was already uploaded in Supplemental File. For baseline characteristics, continuous variables were summarized as mean ± standard deviation as appropriate, and categorical variables were presented as percentage. Quantitative variables were categorized as follows: BMI (kg/m$^2$) <18.5, 18.5–23.9, 24–26.9 and ≥27; 0, 1 and ≥2 hospitalizations for AECOPD; and Charlson Comorbidity Index score ≤2 and >2. As the primary aim of this study was to identify the variables associated with all-cause mortality rather than to test the hypothesis of detecting an expected effect size in a clinical trial, our sample size consideration focused on the size needed to ensure stable and efficient regression coefficients, with at least 6 to 10 events per variable, for example, at least 200 subjects if 20 variables were used in regression models (*Chow, Shao & Wang, 2008*; *Peduzzi et al., 1995*; *Vittinghoff & McCulloch, 2007*). Lung function variables were continuous except for FEV$_1$/height squared using cutoff values of 0.3, 0.4, and 0.5 (*Huang et al., 2018*). Univariate and multivariable Cox proportional hazard regression analyses were performed to estimate crude and adjusted hazard ratios (HRs) of death with 95% CIs. Stepwise Cox proportional hazard regression analysis was performed using candidate variables with $p$ values < 0.35 in univariate analysis in a step-by-step manner (*Chuang, 2022*). Although using candidate variables with $p$ values < 0.157 has been suggested in the literature (*Royston et al., 2009*) using candidate variables with $p$ values < 0.35 would result in many more variables being included. Predictors that are highly correlated with others (*i.e.* a lower $p$-value) contribute little independent information, whereas predictors that are not significant in univariate analysis (*i.e.* a higher $p$-value) should not be excluded as candidates (*Royston et al., 2009*). To assess associations between the significant variables of interest and COPD death during follow-up, Kaplan-Meier survival curves were constructed and compared using the log-rank test. A mortality probability was calculated by applying the logistic regression method to the selected factors. All statistical analyses were performed using SAS software version 9.4. Statistical significance was set at a two-sided $p < 0.05$.

## RESULTS

A total of 14,910 subjects with COPD were obtained from the hospital's database (Fig. 1), of whom 1,280 had complete lung function tests. Of these patients, 456 were retained for analysis after excluding 719 subjects whose spirometry data did not meet the inclusion criteria or met the exclusion criteria, and 105 subjects who had duplicate data or the lung volume measures using the dilution method between January 01, 2010 and December 31, 2012. Between 2013 and 2020 of the study period, the median (IQR) follow-up period was 597 (331–934.5) days. Overall, 81% of the 456 subjects had GOLD stages 2 and 3 (Table 1), and 17.3% and 1.8% had GOLD stages 1 and 4, respectively. The average estV$_D$/V$_{Trest}$ and estV$_D$/V$_{Tpeak}$ values were highly elevated (60.8 ± 5.4% and 44.0 ± 6.4%, respectively), the average RV/TLC value was mildly elevated (137.0 ± 23.6%), and the average D$_L$CO value was mildly reduced (69.5 ± 24.1%). The hospitalized AECOPD rate was 0.60 ± 2.84 PPPY; 13.2% had one hospitalized AECOPD, 10.7% had ≥2 hospitalized AECOPDs, and 76.1% were not hospitalized for AECOPDs (Table 2). Forty-eight subjects (10.5%) died during

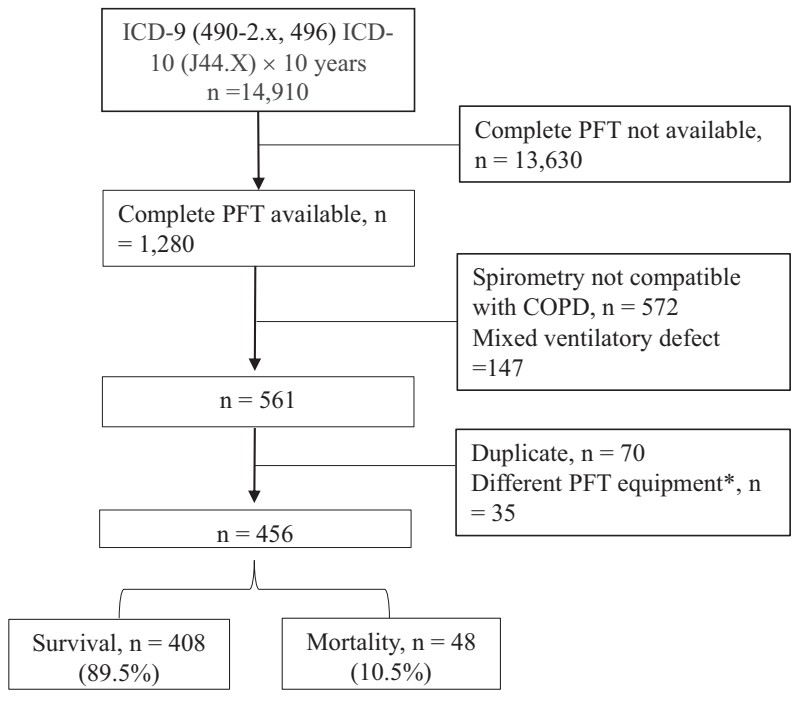

**Figure 1 Flowchart.** Subjects with chronic obstructive pulmonary disease (COPD) were obtained between Jan 1, 2010 and Dec 31, 2020. ICD, international coding of diagnosis; PFT, pulmonary function test. An asterisk (*) indicates that these cases were excluded because single breath dilution method was used to determine lung volumes before 2013, which are significant different from those measured with body plethysmography after 2013.

**Table 1 Demographic characteristics and lung function data (N = 456).**

|  | *N* (%) | Mean ± SD |
|---|---|---|
| Age, years |  | 69.1 ± 10.1 |
| <65 | 157 (34.4) |  |
| ≥65 | 299 (65.6) |  |
| Sex, F:M | 19 (4.2): 437 (95.8) |  |
| Body mass index, kg/m² |  | 24.2 ± 4.1 |
| <18.5 | 31 (6.8) |  |
| 18.5–23.9 | 202 (44.3) |  |
| 24–26.9 | 119 (26.1) |  |
| ≥27 | 104 (22.8) |  |
| Charlson Comorbidity Index |  | 5.3 ± 2.6 |
| ≤3 | 125 (27.4) |  |
| 4–6 | 212 (46.5) |  |
| >6 | 119 (26.1) |  |
| or |  |  |
| ≤2 | 43 (9.4) |  |
| >2 | 413 (90.6) |  |

(Continued)

| | N (%) | Mean ± SD |
|---|---|---|
| estV$_D$/V$_{Trest}$, % | | 60.8 ± 5.4 |
| estV$_D$/V$_{Tpeak}$, % | | 44.0 ± 6.4 |
| FVC% predicted | | 77.6 ± 16.0 |
| FEV$_1$% predicted | | 62.6 ± 18.0 |
| Stage 1, 2, 3, 4 | 79 (17.3), 262 (57.5), 107 (23.5), 8 (1.8) | |
| FEV$_1$/FVC, % | | 61.8 ± 9.5 |
| FEV$_1$/FVC, % [a] | | 62.1 ± 10.0 |
| MMEF% predicted | | 31.4 ± 13.5 |
| PEF% predicted | | 52.4 ± 21.4 |
| IC% predicted | | 78.1 ± 20.8 |
| TLC% predicted | | 98.1 ± 15.2 |
| RV% predicted | | 136.6 ± 39.5 |
| RV/TLC% predicted | | 137.0 ± 23.6 |
| FRC% predicted | | 117.7 ± 27.5 |
| D$_L$CO% predicted | | 69.5 ± 24.1 |
| D$_L$CO/alveolar volume % predicted | | 75.2 ± 25.8 |
| Alveolar volume % predicted | | 74.3 ± 13.8 |

**Notes:**

Abbreviations: D$_L$CO, diffusing capacity of lung for carbon monoxide; estV$_D$/V$_{Trest}$ and estV$_D$/V$_{Tpeak}$, estimated dead space and tidal volume ratios at rest and at peak exercise; FEV$_1$, forced expired volume in one second; FRC, functional residual capacity; FVC, forced vital capacity; IC, inspiratory capacity; MMEF, maximum mid-expiratory flow; PEF, peak expiratory flow; RV, residual volume; TLC, total lung capacity.

**Table 2 The cumulative data during the follow-up period (N = 456).**

| | N (%) | Mean ± SD |
|---|---|---|
| Inhaled medications use | | |
| None | 114 (25.0) | |
| Yes | 342 (75.0) | |
| ICS | 20 (4.4) | |
| LABA | 8 (1.8) | |
| LAMA | 1 (0.2) | |
| LABA+ICS | 227 (49.8) | |
| LABA+LAMA | 7 (1.5) | |
| LABA+LAMA+ICS | 79 (17.3) | |
| Number of hospitalized AECOPD (PPPY) | | 0.60 ± 2.84 |
| 0 | 347 (76.1) | |
| 1 | 60 (13.2) | |
| ≥2 | 49 (10.7) | |
| Cancer | | |
| None | 341 (74.8) | |
| Yes | 115 (25.2) | |
| Mortality | | |

(Continued)

| | N (%) | Mean ± SD |
|---|---|---|
| None | 408 (89.5) | |
| Yes | 48 (10.5) | |
| Cancer | 30 (62.5) | |
| Non-cancer | 18 (37.5) | |
| Acute respiratory failure | 9 | |
| Cardiovascular disease | 3 | |
| Diabetes mellitus-related complications | 2 | |
| Connective tissue disease | 2 | |
| Meningitis | 1 | |
| Others | 1 | |
| Rate @ 1, 2, 3, 4 year, % | 5.2, 11.2, 17.5, 20.1 | |
| Deaths per 100 person-years | | 6.03 (4.54–8.00[*]) |

**Notes:**
AECOPD, acute exacerbation of COPD; ICS, inhaled corticosteroids; LABA/LAMA, long-acting beta bronchodilators/ muscarinic antagonists; PPPY, per person per year.
[*] 95% C.I.

the 7-year study period, including 30 with cancer (median 1.64 years, range 0.1–3.89 years); the 1, 2, 3, and 4-year mortality rates were 5.2%, 11.2%, 17.5%, and 20.1%, respectively. The incidence density of death was 6.03 (95% CI [4.54–8.00]) per 100 person-years. The causes of non-cancer death ($n = 18$) included acute respiratory failure ($n = 9$), cardiovascular disease ($n = 3$), diabetes mellitus-related complications ($n = 2$), connective tissue disease ($n = 2$), meningitis ($n = 1$), and others ($n = 1$).

*Univariate and multivariable analyses.* In Table 3, Model 1 used univariate analysis; Model 2 used multivariable analysis including all variables in Model 1; Model 3 used multivariable analysis including all variables in Model 2 without cancer; Model 4 used multivariable model with stepwise selection. The risk factors for mortality with significant crude HRs were elevated $estV_D/V_{Trest}$ and $estV_D/V_{Tpeak}$, ≥2 hospitalized AECOPDs, advanced age, BMI < 18.5 kg/m$^2$, and cancer; whereas the protective factors were high PEF %, $D_LCO/V_A$%, $V_A$%, and BMI 24–26.9 kg/m$^2$ (Table 3). In multivariable analysis, only cancer remained significant (Table 3). As the mortality rate of patients with cancer with metastasis is speculated to be much higher than that of COPD alone and the study was aimed to investigate the impact of lung physiology on survival, advanced-stage malignancy of any organ was not adjusted in the subsequent analysis. In the Model 3 analysis, paradoxically, $FEV_1$% was a risk factor for mortality, whereas a BMI of 24–26.9 kg/m$^2$ was protective. After stepwise variable selection, $estV_D/V_{Trest}$ and BMI < 18.5 kg/m$^2$ were risk factors for mortality, whereas a BMI of 24–26.9 kg/m$^2$ remained a protective factor (Table 3 and Fig. 2, log-rank, $p < 0.001$). The $estV_D/V_{Trest}$ was subsequently separated into two categories: <61% and ≥61% based on its median value. The categories showed a significant difference in the survival rate (Fig. 2, log-rank, $p = 0.015$).

The mortality probability formula was estimated as follows.

$$p = 1/(1+e^{6.404-7.3 \times estVD/VT}) \text{ for those BMI} < 18.5 \text{ kg/m}^2$$

**Table 3 Univariate and multivariable Cox proportional hazard model analysis for risk of all-cause mortality.**

| | Model 1 | Model 2 | Model 3 | Model 4 |
|---|---|---|---|---|
| | Crude HR (95% C.I.) | Adjusted HR (95% C.I.) | Adjusted HR‖ (95% C.I.) | Adjusted HR (95% C.I.) |
| estV$_D$/V$_{Trest}$, % | 1.07 [1.01–1.13]* | 0.88 [0.65–1.19] | 1.09 [0.94–1.26] | 1.07 [1.01–1.13]* |
| estV$_D$/V$_{Tpeak}$, % | 1.05 [1.00–1.10]* | 1.05 [0.99–1.11] | 1.03 [0.98–1.08] | |
| FVC% | 1.00 [0.98–1.01] | 0.95 [0.77–1.17] | 0.95 [0.87–1.03] | |
| FEV$_1$% | 1.00 [0.99–1.02] | 1.01 [0.81–1.25] | 1.09 [1.01–1.17]* | |
| MMEF% | 1.01 [0.99–1.03] | 1.04 [0.98–1.10] | 1.01 [0.96–1.06] | |
| PEF% | 0.99 [0.97–1.00]* | 0.97 [0.94–1.01] | 0.97 [0.95–1.00] | |
| TLC% | 1.00 [0.98–1.02] | 1.06 [0.80–1.41] | 1.00 [0.93–1.06] | |
| RV/TLC% | 1.00 [0.99–1.02] | 1.09 [1.00–1.20] | 1.01 [0.96–1.05] | |
| FRC% | 1.00 [0.99–1.01] | 1.08 [0.89–1.32] | 1.01 [0.97–1.05] | |
| D$_L$CO% | 0.99 [0.98–1.00] | 1.01 [0.97–1.06] | 1.03 [0.98–1.07] | |
| D$_L$CO/V$_A$% | 0.98 [0.97–1.00]† | 0.97 [0.93–1.02] | 0.96 [0.92–1.00] | |
| V$_A$% | 0.98 [0.96–1.00]* | 0.98 [0.93–1.03] | 0.96 [0.91–1.01] | |
| Age | 1.03 [1.00–1.06]* | 1.04 [0.91–1.18] | 0.94 [0.87–1.02] | |
| Sex | | | | |
| Female | 1 | 1 | 1 | |
| Male | 0.88 [0.21–3.64] | 0.86 [0.05–14.41] | 0.76 [0.12–4.73] | |
| Body mass index, kg/m$^2$ | | | | |
| 18.5–23.9 | 1 | 1 | 1 | 1 |
| <18.5 | 2.68 [1.29–5.57]† | 2.10 [0.71–6.25] | 1.44 [0.60–3.45] | 2.43 [1.15–5.14]* |
| 24–26.9 | 0.34 [0.14–0.83]* | 0.35 [0.12–1.03] | 0.36 [0.14–0.96]* | 0.34 [0.14–0.84]* |
| ≥27 | 0.42 [0.17–1.02] | 0.80 [0.25–2.57] | 0.65 [0.21–1.97] | 0.42 [0.17–1.02] |
| Bronchodilator use | | | | |
| None | 1 | 1 | 1 | |
| Yes | 0.85 [0.45–1.61] | 1.15 [0.51–2.61] | 0.85 [0.41–1.77] | |
| Hospitalized AECOPD | | | | |
| 0 | 1 | 1 | 1 | |
| 1 | 0.58 [0.18–1.91] | 0.31 [0.08–1.15] | 0.35 [0.10–1.23] | |
| ≥2 | 2.50 [1.31–4.77]† | 1.36 [0.57–3.26] | 1.93 [0.83–4.45] | |
| Charlson comorbidity index | | | | |
| ≤2 | 1 | 1 | 1 | |
| >2 | 2.99 [0.72–12.30] | 0.64 [0.11–3.71] | 2.23 [0.43–11.51] | |
| Cancer | | | | |
| None | 1 | 1 | a | a |
| Yes | 5.45 [3.04–9.79]†† | 8.46 [3.91–18.31]†† | a | a |

**Notes:**

AECOPD, acute exacerbation of COPD; D$_L$CO, diffusing capacity of lung for carbon monoxide; estV$_D$/V$_{Trest}$ and estV$_D$/V$_{Tpeak}$, estimated dead space and tidal volume ratios at rest and at peak exercise; FEV$_1$, forced expired volume in 1 s; FRC, functional residual capacity; FVC, forced vital capacity; HR, hazard ratio; MMEF, maximum mid-expiratory flow; PEF, peak expiratory flow; RV, residual volume; TLC, total lung capacity; V$_A$, alveolar volume. ‖ after excluding subjects with cancer. Model 1: Univariate analysis; Model 2: Multivariable analysis including all variables in Model 1; Model 3: Multivariable analysis excluding cancer; Model 4: Multivariable model with stepwise selection (details in statistical methods).

* $p < 0.05$.
† <0.01.
†† <0.0001.
a All variables in the Model 3 analysis were adjusted except cancer.

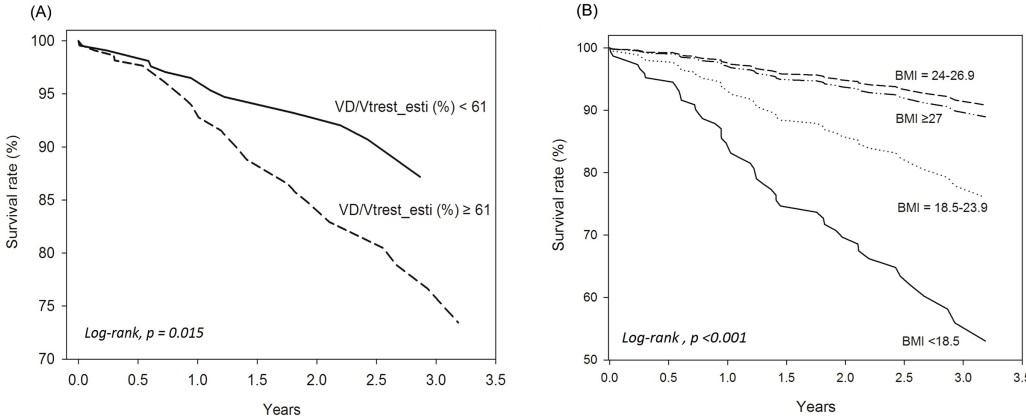

**Figure 2 Cox regression survival analysis was performed according to different values of $V_D$/$V_{Trest\_esti}$ (estimated $V_D$/$V_T$rest) and body mass index (BMI) for all 456 subjects with chronic obstructive pulmonary disease.** (A) est$V_D$/$V_{Trest}$ < 61% is the reference value. (B) Normal BMI (18.5–23.9 kg/m$^2$) is the reference value.

$$p = 1/(1+e^{6.404-7.3\times estVD/VT+1.037\times2}) \text{ for those BMI 24–26.9 kg/m}^2$$
$$p = 1/(1+e^{6.404-7.3\times estVD/VT+0.889\times3}) \text{ for those BMI} \geq 27 \text{ kg/m}^2$$

where est$V_D$/$V_{Trest}$ using the value was counted to the 2nd decimal.

## DISCUSSION

In this study, we found that est$V_D$/$V_{Trest}$ and BMI < 18.5 kg/m$^2$ were risk factors for mortality, whereas a BMI of 24–26.9 kg/m$^2$ was a protective factor. BMI is reported that >21 kg/m$^2$ is a protective factor to survival, whereas ≤21 kg/m$^2$ is detrimental (*Celli et al., 2004*). To the best of our knowledge, this is the first report to link est$V_D$/$V_T$ with the risk of death in patients with COPD without adjusting cancer. A mortality probability formula was generated; however, its utility warrants further studies. The mechanism underlying the predictive ability of est$V_D$/$V_{Trest}$ for COPD death is not clear; however, it may be due to its broad associations with lung and non-lung factors (*Chuang, Hsieh & Lin, 2021*) and is related to hypoxemia and hypercapnia, and thus related to mortality (*Calverley, 2003*; *Dave et al., 2021*; *Mathews et al., 2020*; *Nizet et al., 2005*). Furthermore, measured $V_D$/$V_{Tpeak}$ is a primary pulmonary factor related to exertional dyspnea and then to exercise intolerance (*Chuang, 2022*). The value of est$V_D$/$V_{Trest}$ was extended to COPD mortality prediction in the current study. The discovery of est$V_D$/$V_{Trest}$ in predicting survival among patients with COPD is groundbreaking, suggesting that the application of est$V_D$/$V_{Trest}$ could significantly influence survival predictions.

*Lung function variables*. Although FEV$_1$ (*Leivseth et al., 2013*; *Ou et al., 2014*; *Soriano et al., 2013*), D$_L$CO (*de-Torres et al., 2021*), IC% (*Phillips et al., 2022*) and IC/TLC (*Aalstad et al., 2018*) are the primary pulmonary factors and have been reported to contribute to mortality in patients with COPD, they were not selected in this study. *Huang et al. (2018)* investigated the performance of seven staging methods of FEV$_1$, *i.e.* GOLD, quartiles of FEV$_1$%, z-score of FEV$_1$, quartiles and Miller's cut-off points (FEV$_1$·height$^{-2}$$_{Miller}$, range: 0.3, 0.4, 0.5) of the ratio of FEV$_1$ over height squared, quartiles of the ratio of FEV$_1$·height$^{-3}$ and FEV$_1$ quotient (FEV$_1$Q *i.e.* FEV$_1$ in liters/0.5 L for males; 0.4 L for females) in

predicting outcomes of patients with COPD (*Huang et al., 2018*), and found that staging based on quartile of $FEV_1Q$ was the best predictor, followed by $FEV_1 \cdot height^{-2}_{Miller}$. We tested these two variables in univariate survival analysis and found that the fourth quartile of $FEV_1Q$ (*i.e.* ≥3.82) had a protective effect (0.41 [0.17–0.98], $p = 0.046$), whereas $FEV_1 \cdot height^{-2}_{Miller}$ did not. Although other authors did not find that $FEV_1\%$ was a protective factor (*Martinez et al., 2006*), $FEV_1\%$ was a risk factor for mortality in the Model 3 analysis without adjusting for cancer in this study. This type of paradoxical phenomenon occasionally happens when multiple regression and adjustments are performed. Furthermore, in the present study, $FEV_1\%$ showed no association with $estV_D/V_{Trest}$, consistent with the previous reports (*Chuang, Hsieh & Lin, 2021*). For details on $estV_D/V_{Trest}$, refer to the information provided below.

In contrast, $estV_D/V_{Trest}$, $estV_D/V_{Tpeak}$, PEF%, $D_{LCO}/V_A\%$, and $V_A\%$ were significantly contributing primary lung factors to all-cause mortality in the patients with COPD before adjustment in this study (Table 3). $V_D/V_{Trest}$ is a sophisticated marker of resting gas exchange and is more related to cigarette smoking, carboxyhemoglobin level, pulmonary hypertension and $P_aCO_2$ than $FEV_1\%$ (*Chuang, Hsieh & Lin, 2021*). Moreover, elevated $P_aCO_2$ is related to mortality (*Calverley, 2003*; *Dave et al., 2021*; *Mathews et al., 2020*; *Nizet et al., 2005*). $V_D/V_{Trest}$ can be estimated by cigarette consumption, minute ventilation/$CO_2$ output, arterial oxyhemoglobin saturation, and tidal volume × inspiratory duty cycle (*Chuang, Hsieh & Lin, 2021*). All the reasons mentioned above are likely contributors to the association between $estV_D/V_{Trest}$ and COPD mortality. In this study, we re-derived the $estV_D/V_{Trest}$ prediction equation using lung function variables and demographic data alone. This simplification may allow for the more general use of the prediction equation, even though the predictive power was lower than the previous equation, calculating $estV_D/V_{Trest}$ requires many variables, and the method has yet to be validated. After multivariable Cox regression analysis, $estV_D/V_{Trest}$ remained a significant risk factor (Table 3), however $estV_D/V_{Tpeak}$ did not. This might be because $estV_D/V_{Tpeak}$ and $estV_D/V_{Trest}$ were co-linear ($r = 0.36$, $p = 0.02$).

*BMI*. Reduced BMI is an independent risk factor for COPD and mortality (*Celli, 2010*; *Harik-Khan, Fleg & Wise, 2002*). Even in matched-$FEV_1$, BMI has still been reported to be a marker of COPD phenotypes (*Chuang & Lin, 2014*). COPD patients with a reduced BMI may have a higher rate of impaired peripheral oxygenation (anemia, circulation impairment and deconditioning), where they are taller and more malnourished, anemic and have more hyper-inflation, air-trapping, and diffusion impairment (*Chuang & Lin, 2014*). According to the 10-point BODE index (B: body mass index (BMI), O: obstruction of airflow, D: dyspnea score, E: exercise capacity delineated by 6-min walking distance), BMI > 21 $kg/m^2$ is protective for survival, whereas BMI ≤ 21 $kg/m^2$ is detrimental (*Celli et al., 2004*). Moreover, $FEV_1\%$ and mid-thigh muscle cross sectional area obtained by computed tomography are related to survival (*Nici et al., 2006*), suggesting that fat-free muscle mass is important; however, BMI is easily measured. In this study, the under-weight COPD patients (BMI < 18.5 $kg/m^2$) had a high risk of death (HR of 2.68), the overweight patients (BMI, 24–26.9 $kg/m^2$) had a low risk of death (HR of 0.34), and the obese patients (BMI ≥ 27 $kg/m^2$) had neither effect (Fig. 2). However, obesity was a risk

factor for poor COPD-related outcomes (quality of life, dyspnea, 6-min walking distance, and severe AECOPD) and was dose-dependent (*Lambert et al., 2017*). Nevertheless, BMI is not universally selected in previous studies with multivariable regression analysis (*Soler-Cataluna et al., 2005*).

*AECOPD and co-morbidity.* Some studies have not encompassed AECOPDs or co-morbidities when performing survival analysis. For example, *Martinez et al. (2006)* included many risk variables in a Cox regression model; other studies constructed composite indexes: BODE (*Celli et al., 2004*); DO (D: dyspnea score, O: obstruction of airflow *i.e.* pre-bronchodilator $FEV_1$% or GOLD grade and exertional dyspnea) (*Leivseth et al., 2014*); and ADO (A: age, D: dyspnea score, O: obstruction of airflow) (*Puhan et al., 2009*). However, *Soler-Cataluna et al. (2005)* reported that severe AECOPDs ≥3 episodes were an independent negative impact on prognosis. The authors further integrated exacerbations into the BODE index, where the "E" of BODE was replaced (*Soler-Cataluna et al., 2009*). This omitted the cumbersome 6-min walking test and did not lose power of survival prediction (*Soler-Cataluna et al., 2009*). Although acute respiratory failure accounted for half of the causes of non-cancer death in the current study, acute respiratory failure death only accounted for nine of 109 AECOPD, suggesting that most of the AECOPD were not respiratory failure and the management was effective.
The co-morbidity test (COTE) index has been used to assess the risk of mortality in patients with COPD (*Divo et al., 2012*). Recently, individual diseases such as heart failure and ischemic heart disease rather than CCI score have been shown to have a large effect size on mortality prediction (*Shah et al., 2022*). However, CCI score was used in the current study and stratified as ≤2 or >2. Nevertheless, co-morbidities was also excluded from the COPD all-cause mortality prediction in the previous reports (*Phillips et al., 2022*). Thus, AECOPD and co-morbidities did not contribute to the risk of mortality in multivariable analysis. Even though there are several prediction models, none of them is perfect, and the current study may be helpful to refine future models.

A similar study on hospitalizations and all-cause mortality used logistic regression with stepwise risk stratification (*Groves et al., 2021*), in which demographic and COPD-specific data, and multi-morbidities were used. However, these variables were not sorted clearly, *i.e.* hemoglobin as a COPD-specific variable and anemia as a co-morbidity variable; BMI as a COPD-specific variable. They also included cancer and lung cancer as co-morbidities; however, mortality was best predicted by disease severity (area under the curve (AUC) 0.816; 95% CI [0.805–0.827]), and the predictive ability was only marginally enhanced by adding multi-morbidity indices (AUC 0.829; 95% CI [0.818–0.839]). In contrast, we and other investigators (*Berry & Wise, 2010*; *Huang et al., 2018*) found that cancer was the risk factor highly associated with mortality. In the Model 3 analysis, without adjusting for cancer, BMI and $estV_D/V_{Trest}$ were found to be significantly associated with COPD mortality. The discrepancies across these studies may be due to the use of different variables such as CCI or individual diseases (*Shah et al., 2022*) to score co-morbidities, and differences in the stratification of risk factors (*Groves et al., 2021*). Lastly, composite indexes are generally thought to be better than the primary lung variables for survival prediction. However, some reports do not support this notion, partly because survival

prediction is not their initial purpose for example, GOLD guidelines (*Gedebjerg et al., 2018*; *Lee et al., 2019*; *Leivseth et al., 2013*; *Ou et al., 2014*; *Soriano et al., 2013*).

*Study limitations.* As this was a retrospective study, some variables could not be assessed, such as emphysema (*Martinez et al., 2006*), secondary factors *i.e.* hypoxemia, hypercapnia (*Dave et al., 2021*) and other gas exchange variables, other tertiary factors *i.e.* dyspnea, peak oxygen uptake, frailty (*Lee et al., 2022*; *Nishimura et al., 2002*), and health-related quality of life (*Nishimura et al., 2002*). In this context, composite indexes (*Celli et al., 2004*; *Gedebjerg et al., 2018*; *Lee et al., 2019*; *Ou et al., 2014*; *Puhan et al., 2009*; *Soler-Cataluna et al., 2009*) cannot be calibrated with this dataset. Although chest computed tomography pulmonary angiography (CTPA) may offer detailed diagnostic information in pulmonary arterial hypertension (PAH) (*Condliffe et al., 2023*), chest X-ray (CXR) is more widely used due to its availability (*Mirsadraee et al., 2013*) and lower cost. However, the hila-thoracic ratio (HTR) was not mentioned in the ESC/ERS guidelines for pulmonary hypertension (*Humbert et al., 2022*), despite being first reported by *Chetty, Brown & Light, 1982* and subsequently confirmed by *Mirsadraee et al. (2013)*. As this study is retrospective, CTPA data are not available. Furthermore, the performance of the simplified predictive equations for $estV_D/V_{Trest}$ and $estV_D/V_{Tpeak}$ was deemed insufficient, raising concerns about the accuracy of these predictions. To be noted, all $estV_D/V_{Trest}$ and $estV_D/V_{Tpeak}$ data were derived from the predictive equations rather than direct measurements. Even with direct measurements, there is still a potential for bias, particularly if the subjects exhibit arrhythmias (such as atrial fibrillation) or have NYHA IV heart failure with oscillatory ventilation (*Wasserman et al., 2005*). In addition, COPD-asthma overlap, a phenotype with different outcomes (*Jones et al., 2009*), cannot be fully excluded, even though all of the subjects met the definition of COPD and the diagnostic criteria of the GOLD guidelines, and none of the subjects had a bronchodilator effect in spirometry and were former or current cigarette smokers. Nevertheless, one may still question whether the diagnosis of COPD also included subjects with asthma, as approximately 50% of the enrolled subjects were treated with LABA+ICS and only 1.5% were treated with LABA + LAMA. However, LABA + ICS was advocated for patients with COPD in the GOLD guidelines between 2013 and 2020, and this period overlapped with that of the current study. Being a current smoker is a strong risk factor for mortality in patients with COPD (*Jones et al., 2009*; *Shah et al., 2022*); however, the status of cigarette smoking in some patients was not clear. In addition, as $estV_D/V_{Trest}$ and $estV_D/V_{Tpeak}$ were not measured and their predictive formulae have not been externally validated and only a few patients were included in the study, further studies are needed to verify our results. Moreover, different COPD phenotypes including classical, historical, and new phenotypes may respond to different treatments (*Siafakas, Corlateanu & Fouka, 2017*), potentially introducing bias to the results. Lastly, there may be collection bias concerning the recorded causes of death, as this is a retrospective study.

## CONCLUSIONS

Cancer was the main cause of all-cause mortality in this study; however, $estV_D/V_{Trest}$ and BMI were independent prognostic factors for COPD-related mortality without adjusting

for cancer. $V_D/V_T$ may be a new prognostic factor for COPD and may help to elucidate the relationship between pathogenesis and survival of COPD. Although $V_D/V_{Trest}$ can be estimated using a formula derived from comprehensive pulmonary physiology variables, its clinical implications for survival prediction should be interpreted with caution until the formula has been validated. Formulas for estimating the probability of mortality were proposed; however, further studies are needed to verify their utility.

### Funding
This study received grants from Chung Shan Medical University Hospital (CSH-2021-C-041). The funders had no role in study design, data collection and analysis, decision to publish, or preparation of the manuscript.

### Grant Disclosures
The following grant information was disclosed by the authors:
Chung Shan Medical University Hospital: CSH-2021-C-041.

### Competing Interests
The authors declare that they have no competing interests.

### Author Contributions
- Ming-Lung Chuang conceived and designed the experiments, performed the experiments, analyzed the data, prepared figures and/or tables, authored or reviewed drafts of the article, and approved the final draft.
- Yu Hsun Wang performed the experiments, analyzed the data, prepared figures and/or tables, authored or reviewed drafts of the article, and approved the final draft.
- I-Feng Lin performed the experiments, analyzed the data, authored or reviewed drafts of the article, and approved the final draft.

### Human Ethics
The following information was supplied relating to ethical approvals (*i.e.*, approving body and any reference numbers):

The Institutional Review Board (IRB) of Chung Shan Medical University Hospital (CS2-21018).

### Data Availability
The raw data is available in the Supplemental File.

### Supplemental Information
Supplemental information for this article can be found online at http://dx.doi.org/10.7717/peerj.17081#supplemental-information.

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
