# Peer review of "The contribution of estimated dead space fraction to mortality prediction in patients with chronic obstructive pulmonary disease—a new proposal"

_PeerJ, doi:10.7717/peerj.17081_

## Round 0.1 · original submission · Minor Revisions

Dear authors

Please review and attend to the concerns expressed by the reviewers.

Reviewer 1 ·

Basic reporting

The manuscript fullfils with being clear unambiguous, have a good english, no additional comments
Dead space as main and unique focuse on, intro should be shorten (survival predictors mortality variables and comorbidities); in order to add a bit more why death space fraction should be a good predictor, what reflects, sort of dead space (anatomical, alveolar, physiologic; history about dead space calculation, ……
As on literature, they are well referenced and relevant, but some should be replace by intro modifications
Text structure comforms to perrj standards, disipline norm and has clarity.
Figures and tables are relevant, clear, well labelled explained and described.
All data were entirely supplied

Experimental design

It has original primary research within journal´s scope (medical and health sciences).

Its research question is well defined, relevant & meaningful. ( but considering to leave alone dead space ratio as a main core and taking out bmi for the reasons explained above)

To land a bit more on justification why dead space fills knowledge gap in intro

Has rigorous investigation, aceptable technical and ethical standards.

Its methods described with sufficient detail & information to replicate. (it is clear from various perspectives)

Validity of the findings

I can say impact and novelty not assessed. There is no doubt that rational and benefit is clearly stated, it invites to replication to become a consistent result

All data are provided; also, they are robust, statistically appropiate and treated

Conclusions are well constructed (the correction is tu take aou comments about cancer mortality, it is linked to original research question & limited to supporting results as well.

Additional comments

The title
The contribution of estimated dead space fraction and body mass index to mortality prediction in patients with chronic obstructive pulmonary disease (#92562)
I suggest being modified to:
The contribution of estimated dead space fraction to mortality prediction in patients with chronic obstructive pulmonary disease (#92562)
On basis of your evidence dead space fraction (estvd/vtrest) must be the only one variable mentioned in it, bmi is well known to have an effect on survival, so for me is not necessary to spend more lines on it.

Custom checks
Authors have ethical approval statement and it is appropiate
Manuscript meets article requirements
Manuscript has identifiable info been removed
There are no any experiments or interventions

The manuscript contributes to a better understanding about causes, mechanisms on copd mortality. A different perpective to evaluate survival. I congratulate prof chuang for the previous publications, expertise, and interest in this field.
Please focues this manuscript to only dead space fraction performance as predictor variable, dedicate more lines to explain how it must be relevant to considere it as part of tools to apply for.
Important points to remark:
-no profound criticisms are done, so no more background evidence is necessary other than estvd/vtrest
-no more effort to improve the manuscript is needed. Some observation for intro was done above, in the same way for discussion and conclusion, always focused on dead space fraction
-english language is enough, no corrections is require
-most important issue is predictor mortality= dead space fraction, the next most important issue is to defent why dead space fraction is relevant. Other important thing is to add more information about dead space not shown yet in text
-as we see is just constructive suggestions, no personal opinions are given
-I thank prof chuang et al for submmit manuscript with quality
-strengths: question research, rational, objetive, design (cohort), method, estatistics and results.
-weaknesses: operational definitions for copd (retrospective) and collection bias of the cause of death.

Annotated reviews are not available for download in order to protect the identity of reviewers who chose to remain anonymous.

Reviewer 2 ·

Basic reporting

The topic is relevant since there is little information on physiological variables other than FEV1 that are associated with a higher risk of death in patients with COPD. The English is clear and well-written. The methodology is adequate and the references are appropriate. There are only minor issues that need to be answered.

Experimental design

No comment.

Validity of the findings

No comments

Additional comments

The working paper seems very innovative and helpful for clinical practice because as the authors comment physiology variables different than FEV1 may be more useful. They propose a formula to estimate the probability of mortality in COPD regarding estVD/VTrest.

Some minor observations need to be answered.

The current title is very ambitious. It is suggested to change to: The contribution of estimated dead space mortality prediction in patients with chronic. A new proposal


The following observations arise:

1. Line 34 -38 Regarding crude risk factors and Cox regression analysis, the numbers of these risks should be added.
2. The inclusion of results from the study in lines 90-91 is not appropriate. In a standard research article structure, the Introduction is typically reserved for providing background information, establishing the context of the study, and outlining the research gap. The presentation of results within this section might disrupt the logical flow of the manuscript and could be better placed in the Results section.
3. Line 119 mentions a TLC <80% as an exclusion. Why were they excluded? Was interstitial disease or rib cage defect suspected? It would be prudent to specify this criterion.
4. Line 29-30 you mentioned that the Global Lung Function Initiative reference values (Quanjer et al. 2012) were not utilized, and an alternative approach was adopted to maintain consistency in your lung function reports. Could you please provide additional details on the specific equation or method used to calculate the predicted percentage in your laboratory? Having this information will contribute to a more comprehensive understanding of the methodology and will assist in the evaluation of the reliability and comparability of your.
5. Lines 126-132. In this section, it would be beneficial to include a brief description of the PFTs equipment used for the measurements. Providing details about the specific device and the methodology employed in conducting the PFTs will enhance the transparency and reproducibility of your study.
6. Line 147: The r2 of the simplified VD/VT needs to be higher; the model could imply inaccuracies when making the estimates. How can authors balance accuracy?
7. Line 138: Regarding HTR, radiography measurements are variables that depend on the maximum inspiration of the patients and the operator. Is there a way to replace it and validate it in tomography?. In discussion, it´s important to consider that the validation should use tomography instead of radiography measurements.
8. Line 176: Did the authors also exclude patients already included in the study who died from cancer? If so, doesn't this imply bias?
9. Did the authors have patients with arrhythmias (atrial fibrillation) or NYHA IV heart failure? This could cause variability in the CO2 measurement and, therefore, bias.
10. It could be important to perform a comparison with the formula used to calculate VD/Vt with standard GOLD in this cohort of COPD patients. Please comment it with more details.
11. There is a predictive association with marked mortality in its Kaplan Meier curves. It would have been interesting to compare them with other predictive mortality indices.
12. It would be interesting for the author to specify the r2 AIC and BIC of the models for its prediction since the VD / VT equations on which he is based explain around 30% of the cases with dead space.

---

## Round 0.2 · accepted · Accept

According to the response received, i can confirm that you are attended to the reviewers' comments. I agree with the re-submitted version, and in my opinion such version is ready for publication.

Reviewer 1 ·

Basic reporting

AUTHOR MADE APPROPIATE CORRECTIONS TO THE TITTLE, INTRO, CONCLUSSIONS

Experimental design

NO CHANGES WERE REQUIRED

Validity of the findings

CHANGES TO CONCLUSIONS WERE DONE AS RECOMMENDED

Additional comments

NO MORE COMMENTS
I SATISFIED WITH CORRECTIONS